# Solving Helmholtz Equation with Local Fractional Derivative Operators

**Dumitru Baleanu [1,2,3,\*], Hassan Kamil Jassim [4] and Maysaa Al Qurashi [5]**

[1] Department of Mathematics, Çankaya University, 06530 Ankara, Turkey
[2] Institute of Space Sciences, Magurele, 077125 Bucharest, Romania
[3] Department of Mathematics and Statistics, Faculty of Science, Tshwane University of Technology, Private Bag X680, Pretoria 0001, South Africa
[4] Department of Mathematics, Faculty of Education for Pure Sciences University of Thi-Qar, Nasiriyah 64001, Iraq
[5] Department of Mathematics, College of Science, King Saud University, P. O. BOX 2454, Ryad 11451, Saudia Arabia
[\*] Correspondence: dumitru@cankaya.edu.tr

**Abstract:** The paper presents a new analytical method called the local fractional Laplace variational iteration method (LFLVIM), which is a combination of the local fractional Laplace transform (LFLT) and the local fractional variational iteration method (LFVIM), for solving the two-dimensional Helmholtz and coupled Helmholtz equations with local fractional derivative operators (LFDOs). The operators are taken in the local fractional sense. Two test problems are presented to demonstrate the efficiency and the accuracy of the proposed method. The approximate solutions obtained are compared with the results obtained by the local fractional Laplace decomposition method (LFLDM). The results reveal that the LFLVIM is very effective and convenient to solve linear and nonlinear PDEs.

**Keywords:** coupled Helmholtz equation; local fractional variational iteration method; local fractional Laplace transform (LFLT)

## 1. Introduction

The Helmholtz equation often arises in the study of physical problems involving partial differential equation (PDEs) in both space and time. The Helmholtz equation with local fractional derivative operators in two-dimensional case was suggested in [1,2] as follows:

$$\frac{\partial^{2\alpha} H}{\partial x^{2\alpha}} + \frac{\partial^{2\alpha} H}{\partial y^{2\alpha}} + \omega^{2\alpha} H = f(x, y), \;\; 0 < \alpha \le 1 \tag{1}$$

with the initial value conditions as follows:

$$H(0, y) = \varphi(y), \quad \frac{\partial^{\alpha} H(0, y)}{\partial x^{\alpha}} = \psi(y)$$

where $H(x, y)$ is the unknown function and $f(x, y)$ is a source term. While coupled Helmholtz equations with local fractional derivative operators in two-dimensional case was introduced in [2] as follows:

$$\frac{\partial^{2\alpha} H_1}{\partial x^{2\alpha}} + \frac{\partial^{2\alpha} H_2}{\partial y^{2\alpha}} + \omega_1^{2\alpha} H_1 = f_1,$$

$$\frac{\partial^{2\alpha} H_2}{\partial x^{2\alpha}} + \frac{\partial^{2\alpha} H_1}{\partial y^{2\alpha}} + \omega_2^{2\alpha} H_2 = f_2,$$

(1)

subject to the initial conditions:

$$H_1(0,y) = \varphi_1(y), \frac{\partial^{\alpha} H_1(0,y)}{\partial x^{\alpha}} = \psi_1(y),$$

$$H_2(0,y) = \varphi_2(y), \frac{\partial^{\alpha} H_2(0,y)}{\partial x^{\alpha}} = \psi_2(y).$$

(2)

where $H_1$ and $H_2$ are unknown functions and $f_1(x,y)$ and $f_2(x,y)$ are source terms.

In recent years, many of the approximate and analytical methods have been utilized to solve the PDEs with LFDOs such as the Adomian decomposition method [3–5], variational iteration method [6–11], differential transform method [12,13], series expansion method [14–16], Sumudu transform method [17], Fourier transform method [18], function decomposition method [19,20], Laplace transform method [21,22], reduce differential transform method [23,24], homotopy perturbation Sumudu transform [25], and the existence and uniqueness of solutions for local fractional differential equations [26,27].

The main aim of this work is to propose the local fractional Laplace variational iteration method to solve Helmholtz and coupled Helmholtz equations with LFDOs. It is important to note that the new modification reduces the size of calculations compared to the LFVIM. This paper is organized as follows: In Section 2, the basic mathematical tools of local fractional calculus are introduced. The analysis of the proposed method is given in Section 3. Then in Section 4, the proposed method is implemented to solve some examples. Finally, concluding remarks are presented in Section 5.

## 2. Basic Definitions of Local Fractional Calculus

In this section, we introduce the basic definitions and properties of the local fractional calculus used to describe the proposed schemes.

**Definition 1**. *The local fractional derivative of* $f(x)$ *of order* $\alpha$ *at the point* $x = x_0$ *is given by [19,20,24]:*

$$f^{(\alpha)}(x_0) = \lim_{x \to x_0} \frac{\Delta^{\alpha}(f(x) - f(x_0))}{(x - x_0)^{\alpha}}$$

(5)

where $\Delta^{\alpha}(f(x) - f(x_0)) \cong \Gamma(\alpha + 1)(f(x) - f(x_0)).$

**Definition 2**. *A partition of the interval* $[a,b]$ *is denoted as* $(t_j, t_{j+1}),$ $j = 0,..., N-1,$ *and* $t_N = b$ *with* $\Delta t_j = t_{j+1} - t_j$ *and* $\Delta t = \max\{\Delta t_0, \Delta t_1, .....\}.$ *The local fractional integral of* $f(x)$ *in the interval* $[a,b]$ *is given by [20,24]:*

$$_a I_b^{(\alpha)} f(x) = \frac{1}{\Gamma(1+\alpha)} \int_a^b f(t)(dt)^{\alpha} = \frac{1}{\Gamma(1+\alpha)} \lim_{\Delta t \to 0} \sum_{j=0}^{N-1} f(t_j)(\Delta t_j)^{\alpha}.$$

(6)

If the functions are local fractional continuous then the local fractional derivatives and integrals exist**.**

**Definition 3.** *Let* $\dfrac{1}{\Gamma(1+\alpha)}\int_0^\infty |f(x)|(dx)^\alpha < k < \infty$. *The local fractional Laplace transform of* $f(x)$ *is given by* [19,20]:

$$L_\alpha\{f(x)\} = f_s^{L,\alpha}(s) = \frac{1}{\Gamma(1+\alpha)}\int_0^\infty E_\alpha(-s^\alpha x^\alpha)f(x)(dx)^\alpha,\ 0 < \alpha \le 1 \tag{7}$$

where the latter integral converges and $s^\alpha \in R^\alpha$.

**Definition 4.** *The inverse of the local fractional Laplace transform of* $f(x)$ *is* [19,31]:

$$L_\alpha^{-1}\{f_s^{L,\alpha}(s)\} = f(t) = \frac{1}{(2\pi)^\alpha}\int_{\beta-i\omega}^{\beta+i\omega} E_\alpha(s^\alpha x^\alpha)f_s^{L,\alpha}(s)(ds)^\alpha,\ 0 < \alpha \le 1 \tag{8}$$

where $s^\alpha = \beta^\alpha + i^\alpha\omega^\alpha$, and $\mathrm{Re}(s) = \beta > 0$.

**Theorem 1.** *Suppose that* $L_\alpha\{f(x)\} = f_s^{L,\alpha}(s)$ *and* $L_\alpha\{g(x)\} = g_s^{L,\alpha}(s)$, *then we have the following formulas:*

$$L_\alpha\{af(x)+bg(x)\} = af_s^{L,\alpha}(s) + bg_s^{L,\alpha}(s) \tag{9}$$

$$L_\alpha\{E_\alpha(c^\alpha x^\alpha)f(x)\} = f_s^{L,\alpha}(s-c) \tag{10}$$

$$L_\alpha\{f^{(k\alpha)}(x)\} = s^{k\alpha}f_s^{L,\alpha}(s) - s^{(k-1)\alpha}f(0) - s^{(k-2)\alpha}f^{(\alpha)}(0) - \cdots - f^{((k-1)\alpha)}(0) \tag{11}$$

$$L_\alpha\{E_\alpha(a^\alpha x^\alpha)\} = \frac{1}{s^\alpha - a^\alpha} \tag{12}$$

$$L_\alpha\{x^{k\alpha}\} = \frac{\Gamma(1+k\alpha)}{s^{(k+1)\alpha}} \tag{13}$$

**Proof of Theorem 1:** see [30].

**Definition 5.** *The convolution of two functions is defined symbolically by* [30,31]:

$$\psi_1(x) * \psi_2(x) = \frac{1}{\Gamma(1+\alpha)}\int_0^x \psi_1(t)\psi_2(x-t)(dt)^\alpha \tag{14}$$

or

$$\psi_2(x) * \psi_1(x) = \frac{1}{\Gamma(1+\alpha)}\int_0^x \psi_2(t)\psi_1(x-t)(dt)^\alpha \tag{15}$$

**Theorem 2.** *Let* $L_\alpha\{\psi_1(x)\} = \Psi_{s,1}^{L,\alpha}(s)$ *and* $L_\alpha\{\psi_2(x)\} = \Psi_{s,2}^{L,\alpha}(s)$, *then*

$$L_\alpha\{\psi_1(x) * \psi_2(x)\} = \Psi_{s,1}^{L,\alpha}(s)\,\Psi_{s,2}^{L,\alpha} \tag{16}$$

## 3. Analysis of the Method

In this section, we illustrate the basic idea of the Laplace variational iteration method for the local fractional partial differential equation.

Let us consider the following local fractional partial differential equations:

$$L_\alpha u(x,y) + R_\alpha u(x,y) + N_\alpha u(x,y) = f(x,y), \ 0 < \alpha \le 1 \tag{17}$$

where $L_\alpha = \dfrac{\partial^{k\alpha}}{\partial x^{k\alpha}}$ is the linear LFDO, $R_\alpha$ is a linear LFDO of order less than $L_\alpha$, $N_\alpha$ represents the general nonlinear LFDO, and $f(x,y)$ is the source term.

According to the rule of LFVIM, the correction local fractional functional for Equation (17) is constructed as [6–10,28,29]:

$$u_{n+1}(x) = u_n(x) + {}_0I_x^{(\alpha)}\left(\frac{\lambda(x-\xi)^\alpha}{\Gamma(1+\alpha)}\left[L_\alpha u_n(\xi) + R_\alpha \widetilde{u}_n(\xi) + N_\alpha \widetilde{u}_n(\xi) - f(\xi)\right]\right), \tag{18}$$

where $\dfrac{\lambda(x-\xi)^\alpha}{\Gamma(1+\alpha)}$ is a fractal Lagrange multiplier.

For initial value problems of Equation (17), we can start with:

$$u_0(x) = u(0) + \frac{x^\alpha}{\Gamma(1+\alpha)}u^{(\alpha)}(0) + \cdots + \frac{x^{(k-1)\alpha}}{\Gamma(1+(k-1)\alpha)}u^{((k-1)\alpha)}(0) \tag{19}$$

We now take the Yang-Laplace transform of Equation (18), namely:

$$Ł_\alpha\{u_{n+1}\} = Ł_\alpha\{u_n\} + Ł_\alpha\left\{{}_0I_x^{(\alpha)}\left(\frac{\lambda(x-\xi)^\alpha}{\Gamma(1+\alpha)}\left[L_\alpha u_n(\xi) + R_\alpha \widetilde{u}_n(\xi) + N_\alpha \widetilde{u}_n(\xi) - f(\xi)\right]\right)\right\} \tag{20}$$

or

$$Ł_\alpha\{u_{n+1}\} = Ł_\alpha\{u_n\} + Ł_\alpha\left\{\frac{\lambda(x)^\alpha}{\Gamma(1+\alpha)}\right\}Ł_\alpha\{L_\alpha u_n(x) + R_\alpha \widetilde{u}_n(x) + N_\alpha \widetilde{u}_n(x) - f(x)\} \tag{21}$$

Taking the local fractional variation of Equation (21), which is given by:

$$\delta^\alpha\left(Ł_\alpha\{u_{n+1}\}\right) = \delta^\alpha\left(Ł_\alpha\{u_n\}\right) + \delta^\alpha\left(Ł_\alpha\left\{\frac{\lambda(x)^\alpha}{\Gamma(1+\alpha)}\right\}Ł_\alpha\{L_\alpha u_n(x) + R_\alpha \widetilde{u}_n(x) + N_\alpha \widetilde{u}_n(x) - f(x)\}\right) \tag{22}$$

By using the computation of Equation (22), we get:

$$\delta^\alpha\left(Ł_\alpha\{u_{n+1}\}\right) = \delta^\alpha\left(Ł_\alpha\{u_n\}\right) + Ł_\alpha\left\{\frac{\lambda(x)^\alpha}{\Gamma(1+\alpha)}\right\}\delta^\alpha\left(Ł_\alpha\{L_\alpha u_n(x)\}\right) \tag{23}$$

Hence, from Equation (23) we get:

$$1 + Ł_\alpha\left\{\frac{\lambda(x)^\alpha}{\Gamma(1+\alpha)}\right\}s^{k\alpha} = 0 \tag{24}$$

where:

$$\delta^\alpha\left(Ł_\alpha\{L_\alpha u_n(x)\}\right) = \delta^\alpha\left(s^{k\alpha}Ł_\alpha\{u_n(x)\} - s^{(k-1)\alpha}u_n(0) - \cdots - u_n^{((k-1)\alpha)}(0)\right) = s^{k\alpha}\delta^\alpha\left(Ł_\alpha\{u_n(x)\}\right) \tag{25}$$

Therefore, we get:

$$\textit{Ł}_\alpha \left\{ \frac{\lambda(x)^\alpha}{\Gamma(1+\alpha)} \right\} = -\frac{1}{s^{k\alpha}} \tag{26}$$

Therefore, we have the following iteration algorithm:

$$\textit{Ł}_\alpha\{u_{n+1}\} = \textit{Ł}_\alpha\{u_n\} - \frac{1}{s^{k\alpha}}\textit{Ł}_\alpha\{L_\alpha u_n(x) + R_\alpha u_n(x) + N_\alpha \tilde{u}_n(x) - f(x)\} \tag{27}$$

where the initial value reads as:

$$u_0(x) = u(0) + \frac{x^\alpha}{\Gamma(1+\alpha)} u^{(\alpha)}(0) + \cdots + \frac{x^{(k-1)\alpha}}{\Gamma[1+(k-1)\alpha]} u^{((k-1)\alpha)}(0) \tag{28}$$

Thus, the local fractional series solution of Equation (17) is:

$$u(x,y) = \lim_{n\to\infty} \textit{Ł}_\alpha^{-1}\left(\textit{Ł}_\alpha\{u_n(x,y)\}\right) \tag{29}$$

## 4. Illustrated Examples

In order to illustrate the above results in Section 3, we give the following several examples.

**Example 1.** *Let us consider the following Helmholtz equation involving the local fractional operator:*

$$\frac{\partial^{2\alpha} u(x,y)}{\partial x^{2\alpha}} + \frac{\partial^{2\alpha} u(x,y)}{\partial y^{2\alpha}} = u(x,y), \ 0 < \alpha \le 1 \tag{30}$$

with the initial value conditions as follows:

$$u(0,y) = 0, \ \frac{\partial^\alpha u(0,y)}{\partial x^\alpha} = \cosh_\alpha(y^\alpha) \tag{31}$$

Using relation Equation (27), we structure the iterative relation as:

$$\textit{Ł}_\alpha\{u_{n+1}\} = \textit{Ł}_\alpha\{u_n\} - \frac{1}{s^{2\alpha}}\textit{Ł}_\alpha\left\{ \frac{\partial^{2\alpha} u_n}{\partial x^{2\alpha}} + \frac{\partial^{2\alpha} u_n}{\partial y^{2\alpha}} - u_n \right\}$$

$$= \frac{1}{s^\alpha} u_n(0,y) + \frac{1}{s^{2\alpha}} u_n^{(\alpha)}(0,y) - \frac{1}{s^{2\alpha}}\frac{\partial^{2\alpha}\textit{Ł}_\alpha\{u_n\}}{\partial y^{2\alpha}} + \frac{1}{s^{2\alpha}}\textit{Ł}_\alpha\{u_n\} \tag{32}$$

In view of Equation (28), the initial value reads:

$$u_0(x,y) = u(0,y) + \frac{x^\alpha}{\Gamma(1+\alpha)} u^{(\alpha)}(0,y) = \frac{x^\alpha}{\Gamma(1+\alpha)}\cosh_\alpha(y^\alpha) \tag{33}$$

Making use of Equations (32) and (33), the successive approximate solutions are shown as follows:

$$\textit{Ł}_\alpha\{u_1(x,y)\} = \frac{1}{s^\alpha} u_0(0,y) + \frac{1}{s^{2\alpha}} u_0^{(\alpha)}(0,y) - \frac{1}{s^{2\alpha}}\frac{\partial^{2\alpha}\textit{Ł}_\alpha\{u_0(x,y)\}}{\partial y^{2\alpha}} + \frac{1}{s^{2\alpha}}\textit{Ł}_\alpha\{u_0(x,y)\}$$

$$= \frac{1}{s^{2\alpha}}\cosh_\alpha(y^\alpha) - \frac{1}{s^{4\alpha}}\cosh_\alpha(y^\alpha) + \frac{1}{s^{4\alpha}}\cosh_\alpha(y^\alpha) = \frac{1}{s^{2\alpha}}\cosh_\alpha(y^\alpha), \tag{34}$$

$$\text{\L}_\alpha\{u_2\} = \frac{1}{s^\alpha}u_1(0,y) + \frac{1}{s^{2\alpha}}u_1^{(\alpha)}(0,y) - \frac{1}{s^{2\alpha}}\frac{\partial^{2\alpha}\text{\L}_\alpha\{u_1\}}{\partial y^{2\alpha}} + \frac{1}{s^{2\alpha}}\text{\L}_\alpha\{u_1\}$$

$$= \frac{1}{s^{2\alpha}}\cosh_\alpha(y^\alpha),$$

$$\text{\L}_\alpha\{u_3\} = \frac{1}{s^\alpha}u_2(0,y) + \frac{1}{s^{2\alpha}}u_2^{(\alpha)}(0,y) - \frac{1}{s^{2\alpha}}\frac{\partial^{2\alpha}\text{\L}_\alpha\{u_2\}}{\partial y^{2\alpha}} + \frac{1}{s^{2\alpha}}\text{\L}_\alpha\{u_2\}$$

$$= \frac{1}{s^{2\alpha}}\cosh_\alpha(y^\alpha),$$

$$\text{\L}_\alpha\{u_4\} = \frac{1}{s^\alpha}H_3(0,y) + \frac{1}{s^{2\alpha}}u_3^{(\alpha)}(0,y) - \frac{1}{s^{2\alpha}}\frac{\partial^{2\alpha}\text{\L}_\alpha\{u_3\}}{\partial y^{2\alpha}} + \frac{1}{s^{2\alpha}}\text{\L}_\alpha\{u_3\}$$

$$= \frac{1}{s^{2\alpha}}\cosh_\alpha(y^\alpha),$$

$$\text{\L}_\alpha\{u_n\} = \frac{1}{s^{2\alpha}}\cosh_\alpha(y^\alpha), n \geq 1.$$

Consequently, the local fractional series solution is:

$$u(x,y) = \lim_{n\to\infty}\text{\L}_\alpha^{-1}\left(\frac{1}{s^{2\alpha}}\cosh_\alpha(y^\alpha)\right) = \frac{x^\alpha}{\Gamma(1+\alpha)}\cosh_\alpha(y^\alpha).$$

The result is the same as the one that is obtained by the LFLDM [2].

In Figures 1–3, the three-dimensional plots of the approximate solutions of Equation (30) with initial condition Equation (31) are shown for different values of $\alpha = \frac{1}{2}$, $\ln(2)/\ln(3)$, 1, respectively.

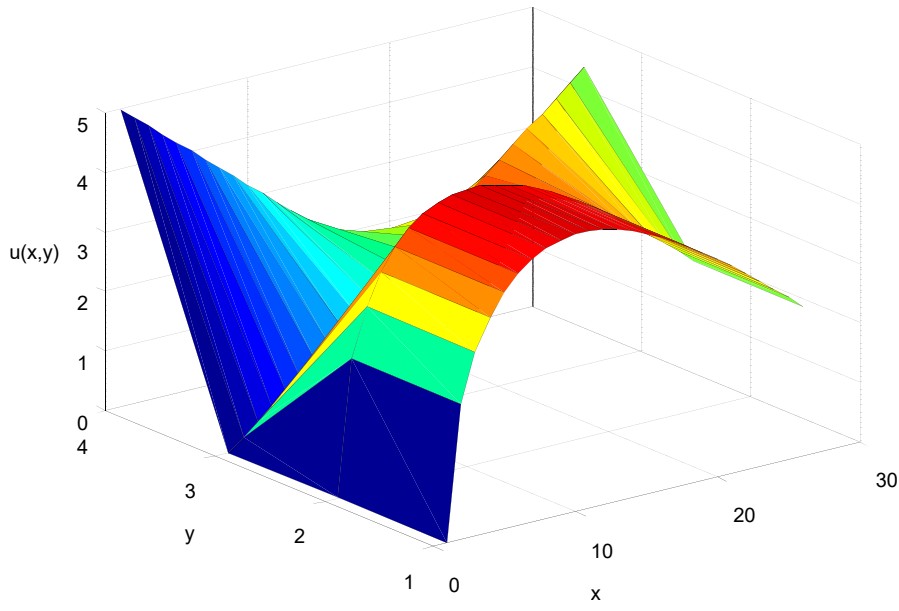

**Figure 1.** The plot of the solution to the local fractional Helmholtz equation with fractional order $\alpha =$ 1/2.

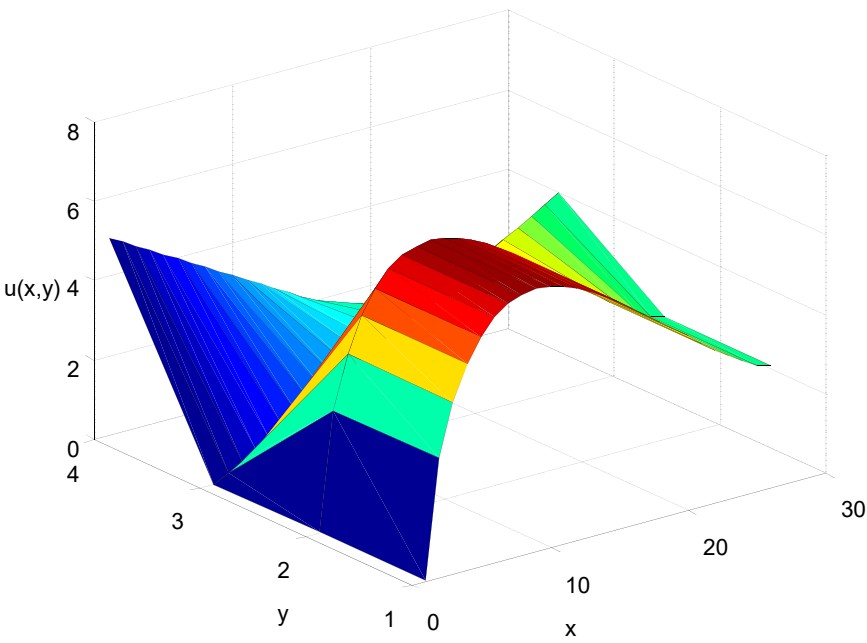

**Figure 2.** The plot of the solution to the local fractional Helmholtz equation with fractional order $\alpha =$ ln(2)/ln(3).

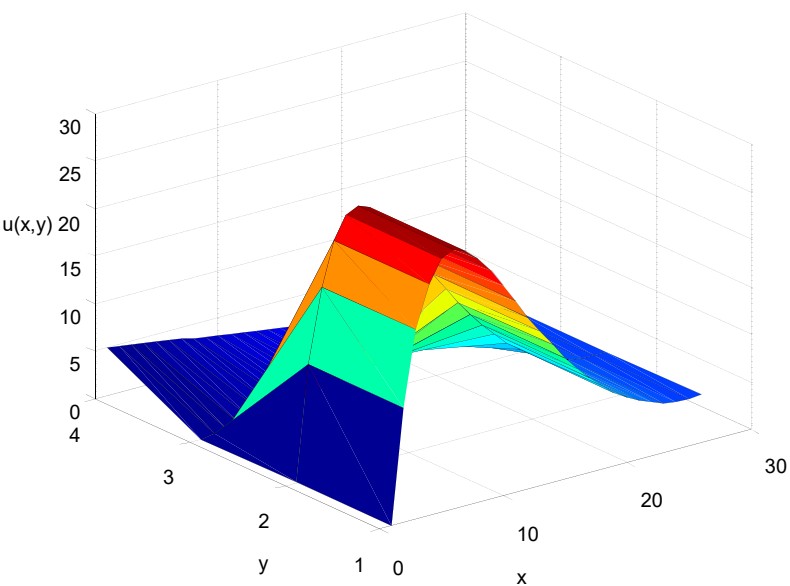

**Figure 3.** The plot of the solution to the local fractional Helmholtz equation with fractional order $\alpha =$ 1.

**Example 2.** *Consider the coupled Helmholtz equations with the local fractional derivative operators:*

$$\frac{\partial^{2\alpha}u(x,y)}{\partial x^{2\alpha}} + \frac{\partial^{2\alpha}v(x,y)}{\partial y^{2\alpha}} - u(x,y) = 0,$$

$$\frac{\partial^{2\alpha}v(x,y)}{\partial x^{2\alpha}} + \frac{\partial^{2\alpha}u(x,y)}{\partial y^{2\alpha}} - v(x,y) = 0,$$
(35)

subject to the initial conditions:

$$u(0,y) = 0, \ \frac{\partial^{\alpha}u(0,y)}{\partial x^{\alpha}} = E_{\alpha}(y^{\alpha}),$$

$$v(0,y) = 0, \ \frac{\partial^{\alpha}v(0,y)}{\partial x^{\alpha}} = -E_{\alpha}(y^{\alpha}).$$
(36)

In view of Equations (27) and (35), the local fractional iteration algorithm can be written as follows:

$$\textit{Ł}_{\alpha}\{u_{m+1}\} = \textit{Ł}_{\alpha}\{u_m\} - \frac{1}{s^{2\alpha}}\textit{Ł}_{\alpha}\left\{\frac{\partial^{2\alpha}u_m}{\partial x^{2\alpha}} + \frac{\partial^{2\alpha}v_m}{\partial y^{2\alpha}} - u_m\right\},$$

$$\textit{Ł}_{\alpha}\{v_{m+1}\} = \textit{Ł}_{\alpha}\{v_m\} - \frac{1}{s^{2\alpha}}\textit{Ł}_{\alpha}\left\{\frac{\partial^{2\alpha}v_m}{\partial x^{2\alpha}} + \frac{\partial^{2\alpha}u_m}{\partial y^{2\alpha}} - v_m\right\},$$
(37)

which leads to:

$$\textit{Ł}_{\alpha}\{u_{m+1}\} = \frac{1}{s^{\alpha}}u_m(0,y) + \frac{1}{s^{2\alpha}}u_m^{(\alpha)}(0,y) - \frac{1}{s^{2\alpha}}\textit{Ł}_{\alpha}\left\{\frac{\partial^{2\alpha}v_m}{\partial y^{2\alpha}} - u_m\right\},$$

$$\textit{Ł}_{\alpha}\{v_{m+1}\} = \frac{1}{s^{\alpha}}v_m(0,y) + \frac{1}{s^{2\alpha}}v_m^{(\alpha)}(0,y) - \frac{1}{s^{2\alpha}}\textit{Ł}_{\alpha}\left\{\frac{\partial^{2\alpha}u_m}{\partial y^{2\alpha}} - v_m\right\},$$
(38)

where the initial value reads:

$$\textit{Ł}_{\alpha}\{u_0(x,y)\} = \textit{Ł}_{\alpha}\left\{\frac{x^{\alpha}}{\Gamma(1+\alpha)}E_{\alpha}(y^{\alpha})\right\} = \frac{E_{\alpha}(y^{\alpha})}{s^{2\alpha}},$$

$$\textit{Ł}_{\alpha}\{v_0(x,y)\} = \textit{Ł}_{\alpha}\left\{-\frac{x^{\alpha}}{\Gamma(1+\alpha)}E_{\alpha}(y^{\alpha})\right\} = -\frac{E_{\alpha}(y^{\alpha})}{s^{2\alpha}}.$$
(39)

Making use of Equations (38) and (39), the successive approximate solutions are shown as follows:

$$\textit{Ł}_{\alpha}\{u_1\} = \frac{1}{s^{\alpha}}u_0(0,y) + \frac{1}{s^{2\alpha}}u_0^{(\alpha)}(0,y) - \frac{1}{s^{2\alpha}}\textit{Ł}_{\alpha}\left\{\frac{\partial^{2\alpha}v_0}{\partial y^{2\alpha}} - u_0\right\},$$

$$\textit{Ł}_{\alpha}\{v_1\} = \frac{1}{s^{\alpha}}v_0(0,y) + \frac{1}{s^{2\alpha}}v_0^{(\alpha)}(0,y) - \frac{1}{s^{2\alpha}}\textit{Ł}_{\alpha}\left\{\frac{\partial^{2\alpha}u_0}{\partial y^{2\alpha}} - v_0\right\},$$
(40)

$$= \frac{E_\alpha(y^\alpha)}{s^{2\alpha}} + \frac{2E_\alpha(y^\alpha)}{s^{4\alpha}},$$

$$= -\frac{E_\alpha(y^\alpha)}{s^{2\alpha}} - \frac{2E_\alpha(y^\alpha)}{s^{2\alpha}}.$$

$$\mathcal{L}_\alpha\{u_2\} = \frac{1}{s^\alpha} u_1(0,y) + \frac{1}{s^{2\alpha}} u_1^{(\alpha)}(0,y) - \frac{1}{s^{2\alpha}} \mathcal{L}_\alpha\left\{ \frac{\partial^{2\alpha} v_1}{\partial y^{2\alpha}} - u_1 \right\},$$

$$\mathcal{L}_\alpha\{v_2\} = \frac{1}{s^\alpha} v_1(0,y) + \frac{1}{s^{2\alpha}} v_1^{(\alpha)}(0,y) - \frac{1}{s^{2\alpha}} \mathcal{L}_\alpha\left\{ \frac{\partial^{2\alpha} u_1}{\partial y^{2\alpha}} - v_1 \right\},$$

$$= \frac{E_\alpha(y^\alpha)}{s^{2\alpha}} + \frac{2E_\alpha(y^\alpha)}{s^{4\alpha}} + \frac{4E_\alpha(y^\alpha)}{s^{6\alpha}},$$

$$= -\frac{E_\alpha(y^\alpha)}{s^{2\alpha}} - \frac{2E_\alpha(y^\alpha)}{s^{4\alpha}} - \frac{4E_\alpha(y^\alpha)}{s^{6\alpha}}.$$

$$\mathcal{L}_\alpha\{u_3\} = \frac{1}{s^\alpha} u_2(0,y) + \frac{1}{s^{2\alpha}} u_2^{(\alpha)}(0,y) - \frac{1}{s^{2\alpha}} \mathcal{L}_\alpha\left\{ \frac{\partial^{2\alpha} v_2}{\partial y^{2\alpha}} - u_2 \right\},$$

$$\mathcal{L}_\alpha\{v_3\} = \frac{1}{s^\alpha} v_2(0,y) + \frac{1}{s^{2\alpha}} v_2^{(\alpha)}(0,y) - \frac{1}{s^{2\alpha}} \mathcal{L}_\alpha\left\{ \frac{\partial^{2\alpha} u_2}{\partial y^{2\alpha}} - v_2 \right\},$$

$$= \frac{E_\alpha(y^\alpha)}{s^{2\alpha}} + \frac{2E_\alpha(y^\alpha)}{s^{4\alpha}} + \frac{4E_\alpha(y^\alpha)}{s^{6\alpha}} + \frac{8E_\alpha(y^\alpha)}{s^{8\alpha}},$$

$$= -\frac{E_\alpha(y^\alpha)}{s^{2\alpha}} - \frac{2E_\alpha(y^\alpha)}{s^{4\alpha}} - \frac{4E_\alpha(y^\alpha)}{s^{6\alpha}} - \frac{8E_\alpha(y^\alpha)}{s^{8\alpha}}.$$

$$\mathcal{L}_\alpha\{u_m\} = \sum_{k=0}^{m} \frac{2^k E_\alpha(y^\alpha)}{s^{(2k+2)\alpha}},$$

$$\mathcal{L}_\alpha\{v_m\} = -\sum_{k=0}^{m} \frac{2^k E_\alpha(y^\alpha)}{s^{(2k+2)\alpha}}.$$

Consequently, the local fractional series solution is:

$$u = \mathcal{L}_\alpha^{-1}\left( \frac{E_\alpha(y^\alpha)}{s^{2\alpha}} + \frac{2E_\alpha(y^\alpha)}{s^{4\alpha}} + \frac{4E_\alpha(y^\alpha)}{s^{6\alpha}} + \cdots \right)$$

$$v = \mathcal{L}_\alpha^{-1}\left( -\frac{E_\alpha(y^\alpha)}{s^{2\alpha}} - \frac{2E_\alpha(y^\alpha)}{s^{4\alpha}} - \frac{4E_\alpha(y^\alpha)}{s^{6\alpha}} - \cdots \right) \tag{40}$$

$$= E_\alpha(y^\alpha)\left( \frac{x^\alpha}{\Gamma(1+\alpha)} + \frac{2x^{3\alpha}}{\Gamma(1+3\alpha)} + \frac{4x^{5\alpha}}{\Gamma(1+5\alpha)} - \cdots \right) = E_\alpha(y^\alpha)\frac{\sinh_\alpha\left(\sqrt{2}x^\alpha\right)}{\sqrt{2}},$$

$$= -E_\alpha(y^\alpha)\left( \frac{x^\alpha}{\Gamma(1+\alpha)} + \frac{2x^{3\alpha}}{\Gamma(1+3\alpha)} + \frac{4x^{5\alpha}}{\Gamma(1+5\alpha)} - \cdots \right) = -E_\alpha(y^\alpha)\frac{\sinh_\alpha\left(\sqrt{2}x^\alpha\right)}{\sqrt{2}}.$$

The result is the same as the one that is obtained by the LFLDM [2] and LFLHPM [30].

In Figures 4 and 5, the three-dimensional plots of the approximate solutions of Equation (35) with initial condition Equation (36) are shown for different values of $\alpha = \frac{1}{2}$, 1, respectively.

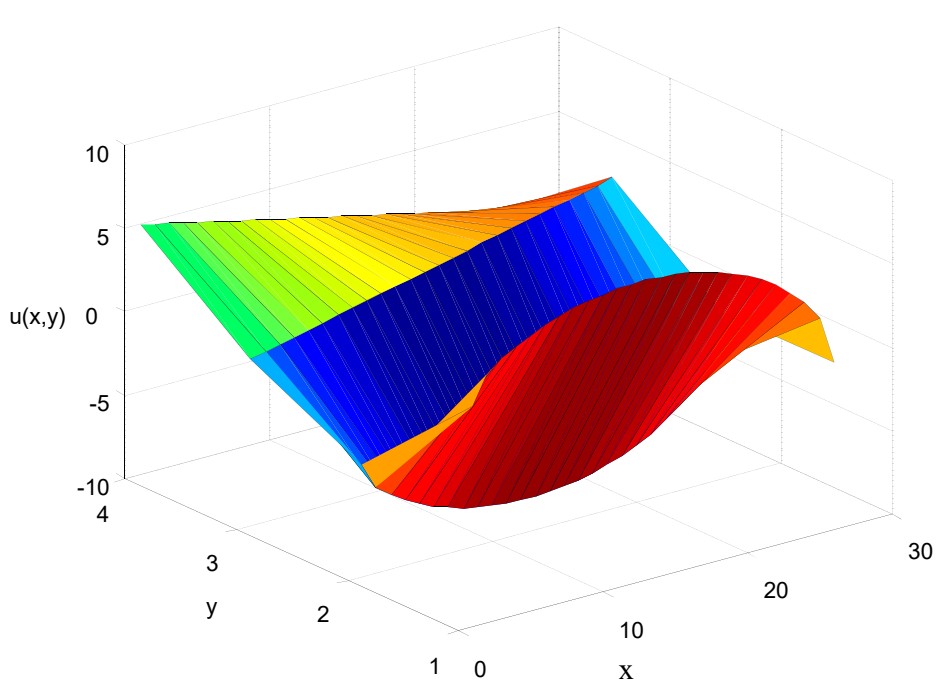

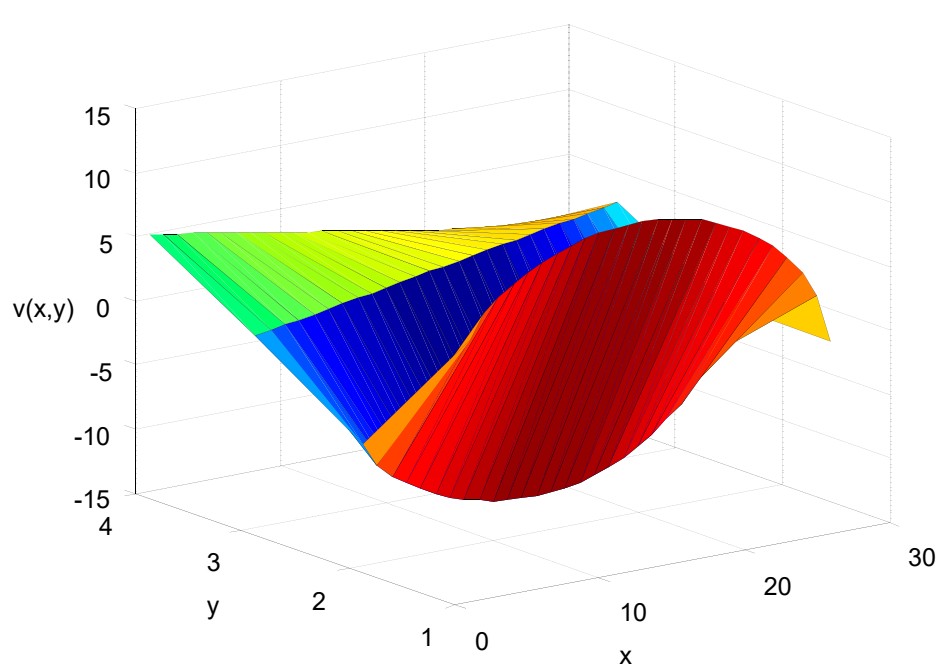

**Figure 4.** The plot of the solutions to the coupled Helmholtz equations involving the local fractional derivative operator with fractional order $\alpha = 1/2$.

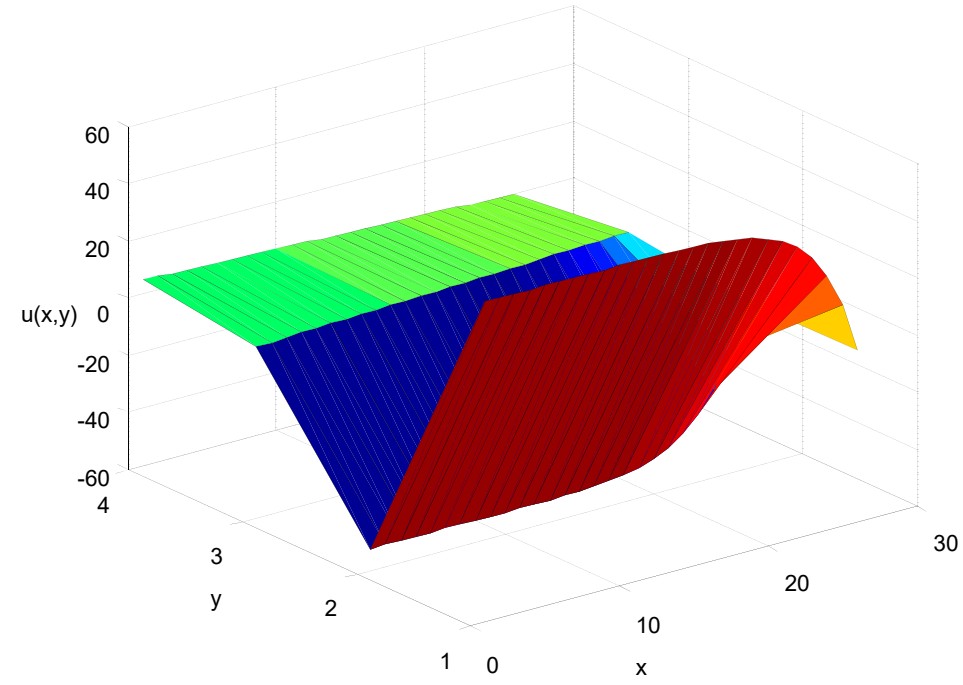

.

**Figure 5.** The plot of the solutions to the coupled Helmholtz equations involving the local fractional derivative operator with fractional order $\alpha = 1$.

## 5. Conclusions

In this work we utilized the coupling method of the local fractional variational iteration method and Laplace transform to solve Helmholtz and coupled Helmholtz equations and their approximate solutions were obtained. The local fractional Laplace variational iteration method was proved to be an effective approach for solving partial differential equations with local fractional derivative operators due to the excellent agreement between the obtained approximate solution and the exact solution. A comparison was made to show that the method has a small size of computation in comparison with the computational size required in other numerical methods, and its rapid convergence shows that the method is reliable and introduces a significant improvement in solving linear and nonlinear partial differential equations with local fractional derivative operators.

**Author Contributions:** H.K.J. wrote some sections of the manuscript; D.B. and M.A. prepared some other sections of the paper and performed analyses. All authors have read and approved the final manuscript.

**Funding:** This research received no external funding.

**Acknowledgments:** The authors are very grateful to the referees for useful comments and suggestions towards the improvement of this paper.

**Conflicts of Interest:** The authors declare no conflict of interest.

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
