# Peer review of "Solving Helmholtz Equation with Local Fractional Derivative Operators"

_fractalfract, doi:10.3390/fractalfract3030043_

Round 1
Reviewer 1 Report
The paper presents a new analytical technique based on the local fractional Laplace Transform and the local fractional variational iteration method. The method is applied to different Helmholtz equations, one partial differential equation and one coupled partial differential equation.
The main ideas of the paper are well presented. The sentences are objective and correctly linked. The vocabulary is adequate and technical.
The foundation works are adequately cited. The simulations are a good addition as they enhance the understanding of the behavior of the solutions.
Nonetheless, there are some important points that should be taken into consideration as they severely hinder the quality of the paper overall.
a) There are grammatical issues that should be corrected during a general revision of the text. Lines 153-160 and 198-205, however, should be rewritten, as there are a considerable amount of issues including typos, grammatical errors and inadequate spacing.
b) The alignment of the equations throughout the paper should also be checked, as some of them are misaligned (special attention should be paid to the equation numbers alignment).
c) There are some minor problems regarding Figures 1, 2, 3, 5, 6 and 7 of the paper. The axis of all figures should be labeled. Additionally, there are grammatical and spacing issues in the label of all these figures.
d) There are some major problems regarding Figures 4 and 8 and they should be entirely remade as they not fit the standards of publication. The labels, legends and markers are too small, the lines are too thin and general understanding of the plot is heavily impaired.
All in all, the paper is interesting. It is important that the issues aforementioned should be corrected before publication.
Reviewer 2 Report
The paper needs major revisions. Please see the attached file.

Round 2
Reviewer 2 Report
The authors have improved their paper, and I recommend it for publication.